# Seed Priming with Zinc Oxide Nanoparticles to Enhance Crop Tolerance to Environmental Stresses

**DOI:** 10.3390/ijms242417612

**Published:** 2023-12-18

**Authors:** Domenica Tommasa Donia, Marilena Carbone

**Affiliations:** Department of Chemical Science and Technologies, University of Rome Tor Vergata, 00133 Roma, Italy; dodomemy@gmail.com

**Keywords:** seed priming, ZnO NPS, crops, stress alleviation

## Abstract

Drastic climate changes over the years have triggered environmental challenges for wild plants and crops due to fluctuating weather patterns worldwide. This has caused different types of stressors, responsible for a decrease in plant life and biological productivity, with consequent food shortages, especially in areas under threat of desertification. Nanotechnology-based approaches have great potential in mitigating environmental stressors, thus fostering a sustainable agriculture. Zinc oxide nanoparticles (ZnO NPs) have demonstrated to be biostimulants as well as remedies to both environmental and biotic stresses. Their administration in the early sowing stages, i.e., seed priming, proved to be effective in improving germination rate, seedling and plant growth and in ameliorating the indicators of plants’ well-being. Seed nano-priming acts through several mechanisms such as enhanced nutrients uptake, improved antioxidant properties, ROS accumulation and lipid peroxidation. The target for seed priming by ZnO NPs is mostly crops of large consumption or staple food, in order to meet the increased needs of a growing population and the net drop of global crop frequency, due to climate changes and soil contaminations. The current review focuses on the most recent low-cost, low-sized ZnO NPs employed for seed nano-priming, to alleviate abiotic and biotic stresses, mitigate the negative effects of improper storage and biostimulate plants’ growth and well-being. Taking into account that there is large variability among ZnO NPs and that their chemico-physical properties may play a role in determining the efficacy of nano-priming, for all examined cases, it is reported whether the ZnO NPs are commercial or lab prepared. In the latter cases, the preparation conditions are described, along with structural and morphological characterizations. Under these premises, future perspectives and challenges are discussed in relation to structural properties and the possibility of ZnO NPs engineering.

## 1. Introduction

The 2030 Agenda of sustainable development goals of the United Nations foresees the “Zero hunger” pursuit as its second point. It is estimated that in 2022, approximately 735 million people—or 9.2% of the world’s population—found themselves in a state of chronic hunger. Food insecurity and hunger are global issues, exacerbated by climate changes triggering different types of stressors [1]. The worsening of climate conditions constantly affects crops yields and quality, causing food shortages worldwide, especially in areas under threat of desertification. A combination of climate stressors, such as drought and heat, significantly impacts crops yield by decreasing harvest index, shortening crop life cycle, and altering seed number, size and composition [2]. A robust negative association is projected among warming, caloric yield and crop frequency. By the 2050s, it is foreseen that crop frequency augmentation in cold regions will be offset by larger decreases in warm regions, resulting in a net global crop frequency reduction (−4.2 ± 2.5% in high-emission scenario), suggesting that climate-driven decline will exacerbate crop production loss [3]. Soil degradation and poor water retention lead to an imbalance between food and feed production as well as carbon storage in the ecosystem. On a larger scale, micronutrient deficiency (or hidden hunger) lead to soil erosions and nutrient runoffs, causing soil infertility, affecting human beings through malnutrition and other related diseases [4,5]. On the other hand, heavy metals from various anthropogenic sources are dumped into the environment, causing accumulation in soil and posing a severe threat for soil, plants, food security and human health. Heavy metals are persistent and non-degradable, and above threshold levels, they alter the soil–plant systems by disrupting physiological and biochemical metabolisms [6]. Therefore, contamination of soil by heavy metals, especially those used for the cultivation of food/feed crops, is a serious environmental and agronomical issue [7].

A major role in holding the pledge against the environmental stressors induced by climate changes is played by nanomaterials, since they can be employed with several functions. They are more efficient than traditional fertilizers, with a site-specific and controlled release of nutrients that increases the efficacy of plant uptake and reduces negative environmental effects related to nutrients loss into the environmental matrices. In addition, they offer a great contribution to enhancing agricultural productivity via gradual release and proper pesticide dosage to aid pest control [8]. The administration of nanomaterials to food crops may help to readily supply nanoscale nutrients, strengthening crops, contributing to food crops biofortification and improving the nutritional quantity and quality of foods [9]. In addition, it may enhance the defense potential against exposure to toxic concentrations of heavy metals resulting in a higher capability for plants to survive and develop under stress conditions [10].

The application of nanomaterials at various phases of plant growth is labelled as priming and is considered a form external stimulus or short-lasting preconditioning that may be applied to several compartments with the purpose of enhancing physiological and biochemical mechanisms of defense, thus promoting earlier, faster and stronger responses to stresses [11,12]. Priming may be applied to seedlings or their parts, as well as young plants in active growth phases, allowing the amelioration/evaluation of defense responses in specific plant regions or organs (for instance, the root system [13,14] or the foliar district [15]). However, the most frequent application is seed priming, a pre-sowing treatment, before the planting phase, aimed at physiological changes in the seeds that serve as a germination synchronizer with numerous advantages from an agricultural viewpoint [16,17]. Seed priming may reduce dormancy by the induction of physiological changes before planting and enhance the growth and yield of treated plants. In addition, it may mitigate the negative effects of long seed storage by counteracting the alterations of biochemical properties responsible for low seed vigor and diminished capability of enacting protection mechanisms against pathogens [18]. Different seed priming solutions were proposed to boost plant germination, including phyto- and biopriming [19], and hydropriming, halopriming, osmopriming, hormopriming and nano-priming [20]. Seed nano-priming, i.e., priming with solutions containing nanoparticles (NPs) [21], is an innovative, promising, simple and cost-effective strategy to improve plant growth and development, tolerance to biotic and abiotic stresses, and the production capacity of crops [19,22,23]. Nano-priming is a considerably more effective method compared to all other types of seed priming, its efficacy being based on the enhanced capability of seeds to rapidly absorb nutrients and renovate seed metabolism [20]. As nanoscale materials have versatile physicochemical properties and a large surface/volume ratio, they are expected to be better absorbed by seeds as compared to bulk chemicals, thus triggering enhanced molecular interactions at the cellular level. The flexibility in altering the surface chemical properties of nanomaterials can facilitate better interaction with the seeds/seedlings while inhibiting the wastage of priming agents. Their action is displayed by promoting the plants’ production of phytohormones, of antioxidant enzymic and non-enzymic molecules, and by inducing the overexpression of new water channels [9]. In addition, several studies have demonstrated their bacteriostatic effect towards a wide panel of pathogenic and non-pathogenic bacteria and fungi [24]. The striking features of nanoparticles are connected to the possibility of developing electron exchange and enhancing the surface reaction of various components of plant cells and tissues. Nano-priming is responsible for the formation of nanopores in shoot, fostering water uptake. It activates reactive oxygen species (ROS), as well as antioxidant mechanisms in seeds, thus stimulating fast hydrolysis of starch. It also induces the expression of aquaporin genes that are involved in the intake of water and mediates H_2_O_2_, or ROS, dispersed over biological membranes.

Nano-priming by metal oxide nanoparticles (MO NPs), and especially by ZnO NPs, has largely increased in recent years, also on account of several methods implemented for their achievement and detection [25,26,27,28,29,30,31,32,33,34,35,36,37,38].

In general, ZnO NPs are widely applied in agriculture and food industries, and can be used as fertilizers and micronutrients to improve yield and food quality, especially when crops are grown in zinc-poor soils. Initially, ZnO NPs priming was put in place for biofortification in response to Zn deficiency in soil and consequent poor dietary intake [39]. Comparison with routine supplements indicated better plant development with nano-priming [40]. ZnO NPs were applied through several pathways, including the foliar district of seedling and plants, though seed priming became the primary choice due to its greater efficacy. Since high seed Zn content has a starter fertilizer effect, it is responsible for achieving good crop yield and improvements in rice grain. The concentration of Zn in plants is a result of enhanced Zn uptake by roots after flowering. Therefore, supplementary Zn by ZnO NPs application became important for improving both grain yield and grain Zn content [41,42]. Further developments foresaw the investigation of the beneficial effects of stress alleviation [43].

It must be added that the benefits of ZnO NPs priming are very much dose dependent and need to be offset against the shortcomings of too high a dose, in terms of inhibition of seed germination and root growth as well as retarded growth of seedlings. In particular, a dose of 1000 mgL^−1^ is considered the threshold level, beyond which, significant changes occur in the activity of soil enzymes, such as the inhibition of protease, catalase and peroxidase, which are bioindicators of soil quality and health [44,45,46,47].

Since the efficacy of ZnO NPs is related to their physico-chemical properties, their synthesis and characterization are pivotal for a correct assessment of the priming–benefits correlation. However, these aspects are not always fully explored in seed priming investigations, with the focus mostly on growth promotion and the assessment of defense mechanisms.

This review briefly illustrates the priming mechanisms, the types of crops primed with ZnO NPs as well as the most recently published studies using ZnO NPs as tools against abiotic and biotic stresses. In particular, the action was analyzed against draught, salinity, accumulation of heavy metals and arsenic, poor storage conditions and biotic stress for the amelioration of the most common crops. Finally, ZnO NPs priming for improving plant growth is also evaluated from the perspective of targeted engineering via suited coating.

## 2. Seed Nano-Priming Mechanism NPs Entry

Seed priming by ZnO NPs may alleviate plants’ abiotic and biotic stress, act as a biostimulant causing an increase in germination rate, seedling and plant growth and overall fresh weight, and improve the biomass and photosynthetic machinery.

The effects of seed priming are implemented through several mechanisms. NPs affect the germination and vigor of plants by the stimulation and improvement of seed metabolic rate, vigor index and seedling characteristics [48], especially in the case of ZnO NPs priming [49]. Furthermore, seed nano-priming plays a role in nutrient uptake. This is particularly important, since poor nutrient uptake negatively affects the whole growth process, including root formation, seedling growth, flowering, and fruit formation [50,51]. Implementing nutrient delivery by conventional management systems may not be efficient, thus opening the way to nano-priming technology [52,53]. In fact, NPs may improve nutrient uptake by altering the metabolism of seeds, for instance, by a steep rise in α-amylase activity via gibberellic acid [54], the breakdown of stored starch, and stimulation of the release of growth regulators, ultimately fostering the growth and productivity of plants. This can also be monitored by the activity of enzymes like α-amylase, which has a steep increase upon seeding.

Management of the antioxidant system is crucial to a plant’s well-being, since it prevents potential cellular molecule damage by maintaining ROS homeostasis [55]. Enzymatic and non-enzymatic antioxidant enzymes participate in the ROS management system: catalase (CAT), peroxidase (POX), ascorbate-peroxidase (APX), superoxide-dismutase (SOD), phenylalanine ammonia lyase (PAL), glutathione, and glutathione reductase. Seed priming affects the antioxidant enzyme system of plants at several levels.

It was observed that the activity of antioxidant enzymes increased at various degrees, depending on the type of plant and on the specific priming, though the exact action pathway was not quite fully disclosed. Some correlations were hypothesized, though. For instance, H_2_O_2_ radicals were significantly reduced in tomato, cucumber, and pea nano-primed seeds due to increased SOD and CAT activity [56,57,58]. On the other hand, H_2_O_2_ is responsible for carbonylating proteins, initiating and changing kinase transduction pathways; therefore, it can interfere with the expression of multiple genes in the germination process [59]. Seed nano-priming plays a role in ROS and lipid peroxide regulation. Seeds accumulated in the seed coat induce ROS accumulation and activate a number of downstream processes [60]. ROS are produced in plants’ chloroplasts, mitochondria, and peroxisomes as a by-product during aerobic metabolism [61,62]. They may irreversibly damage DNA, but they also act as signaling molecules that allow plants to grow normally and mitigate abiotic stresses [63,64]. Increased ROS accumulation in plant cells upon seed nano-priming helps to break the bonds among the polysaccharides in the cell wall of seed endosperm, thus facilitating quick and healthy seed germination [65]. It triggers processes of breaking seed dormancy and stimulating seed germination [66] through the activation of the synthesis of gibberellic and abscisic acid [67,68]. The lowering of ROS by NPs may result in a large rise in antioxidant enzyme levels. This may help increasing tolerance towards stress factors, such as salinity and drought [69,70]. Heavy metals affect crop growth development and production, hampering plant growth progressions. However, in general, NPs regulate plant physiological and biochemical parameters to reduce their detrimental effects [71]. Several studies indicate a connection between NPs priming and decreases in toxic metal accumulations by stimulating antioxidant enzyme activities and decreasing ROS and lipid peroxidation [72]. A scheme of many nano-priming mechanisms is summarized in Figure 1.

Upon seed priming, the entry of the NPs is proposed by three mechanisms [73]. In the first one, NPs are considered as molecules crossing the plasma membrane by a direct diffusion process, the efficacy depending on the NPs’ properties, i.e., the size, hydrophobicity, constitution, charge, and shape of the particles [74]. In the second mechanism, NPs are actively transported into the cell by endocytosis [75]. The third mechanism occurs through transmembrane proteins or through channels that regulate the movement of NPs into cells [76]. However, the actual limiting factors for NP entrance are mostly related to their high degree of specificity, the capability of generating a “least open” pathway, and the presence of small pores [77].

## 3. Application to Different Crops

Nano-priming techniques are deemed beneficial for improving crop yields, frequencies and seedling and plant well-being. In this framework, great effort has been devoted to promote and support nanotechnologies in agriculture under different flagships such as the Nano Mission [78] of the Indian government to encourage private-sector investment and support the growth and commercial application of nanotechnology. Broad, systematic and low-cost applications of NPs on fields are general requirements for sustainable agricultural stimulation. The success of seed nano-priming as a form of biostimulation and an agent of contrast against abiotic and stresses varies based on the species of plants and the characteristics of treating agents. However, ZnO NPs appear to be very efficient and versatile among nanoparticles, for seed nano-priming purposes, since they are suitable for targeting different types of crops worldwide. Besides this, their production is low cost and can be achieved with environmentally friendly methods [79]. Crops which were successfully tested for ZnO NPs nano-priming include produce grown in localized areas, as well as global harvests. Large consumption cereals are a primary target for improved production, in dry, salty and contaminated soils, especially since cereal crops play an important role in satisfying daily calorie intake in the developing world. However, the Zn concentration in grain is inherently very low, particularly when grown on Zn-deficient soils. Rice alone (*Oryza sativa* L.) is one of the major staples, feeding more than half of the world’s population. It is grown in more than 100 countries, predominantly in Asia [80]. Hence, rice, wheat, maize, sorghum, chickpeas and several varieties of faba beans (lupine, *vigna mungo*, *vigna radiata* and *vicia faba*) were recently probed for seed priming with ZnO NPs. However, tests are also being performed on different types of edibles, such as spinach (*basella alba*) and tomatoes. Finally, textile fibers (cotton) and vegetables for animal feed and industrial oil production (rapeseed, *Brassica napus* L.) were also investigated for ZnO NPs seed priming.

## 4. ZnO NPs Seed Priming against Abiotic Stresses

The efficacy of seed priming is dose dependent, and investigations may be carried out in vitro, in pot or in field. In the following sections, a summary of the latest research is reported on the effects of ZnO NPs priming on the most widespread crops worldwide. Drought, soil salinity and heavy metal and arsenic accumulation are listed among abiotic stresses as well as improper seed storage. In addition, action against biotic stresses is considered in association with biotic remedies. The sheer biostimulating effect of ZnO NPs is also reported. Details are mentioned for the major effects, doses and mechanisms when specified in the original papers. The ZnO NPs used in the experiments were of two types, commercial or purposely synthetized. Information on ZnO NPs preparation and characterization methods was reported where available. Figure 2 summarizes the main actions of ZnO NPs seed priming and the consequent effects on crops. The type of stress, ZnO NPs doses, target plants and major benefits of seed priming are reported in Table 1, whereas the type of ZnO NPs and the synthesis and characterization methods (if available) are reported in Table 2.

### 4.1. Drought Stress

Draught is a major abiotic stress, regularly associated with excessive heat and high salt levels that affects more than one third of croplands in the world [20]. Water deficit seriously threatens plants’ metabolism, compromising crops’ quality and quantity as well as their nutritional power, thus affecting the dietary balance in the food and feed chain. Wheat (*Triticum aestivum* L.) and rice (*Oryza sativa* L.) are the major cereals cultivated as a source of caloric nutrient [102]. Zinc has an important role in cereal growth; since it is an essential nutrient for normal homeostasis, it is actively involved in the metabolism of proteins, the synthesis of hormones and enzymes and it helps to alleviate oxidative damages, thus improving nutrient profiles [22,103]. Seed priming with ZnO NPs plays a role in alleviating drought-induced damages in cultivars grown in environments subjected to water shortage [81,82]. This was achieved, for instance, by Mazhar et al. [81], who performed seed nano-priming by dipping the rice seeds in suspensions of ZnO NPs at different concentrations (0, 5, 10, 15, 25 and 50 ppm) for 24 h. Afterwards, primed seeds were planted in water shortage conditions, corresponding to 35% irrigation as compared to normal pots. The agronomic profile of rice from ZnO NPs-primed seeds increased due to the activation of plant internal defense mechanisms to counteract abiotic stress. For this investigation, commercial ZnO NPs were employed with an average size of 20–30 nm, whose characterization was supplied by the provider. Priming of wheat seeds by ZnO NPs had different effects on the growth and well-being of plants depending on the dose. In an investigation by Abbas et al. [82], wheat seeds were treated with commercial ZnO NPs for 12 h at 40, 80, 120 and 160 ppm, sowed, and grown in pots both under water stress and well-watered conditions. After 4 weeks, all primed plants displayed better biometric parameters than the un-primed counterparts, under both watering conditions. In particular, seed exposure to 120 ppm ZnO NPs resulted in higher chlorophyll content, a larger amount of plant nutrients and enhanced antioxidant activity. Also, in this case, the only information on the ZnO NPs’ chemico-physical properties was available from the supplier. A field trial was conducted by El-Bassiouny et al. [83] through two consecutive seasons of wheat farming under different water irrigation requirements. The investigation showed that drought stress decreases morphological parameters, photosynthetic pigments and wheat yield, whereas total soluble sugars, total free amino acids, proline content and water productivity increased. The tolerance to drought varied with wheat cultivars. In particular, seed priming with 10 mg/L commercial ZnO NPs in trial fields neutralized the negative influence of water deficit, stimulated the appearance of some proteins and showed no DNA damage. No data are available, neither on the production methods of the ZnO NPs, nor on their chemico-physical characteristics.

### 4.2. Salt Stress

Salinity is an abiotic stress, among the major threats to sustainable crop production in arid and semiarid lands [104], responsible for more than 50% of crop losses worldwide [84]. Therefore, in an attempt to contain this damage, ZnO NPs have been employed in test assays as trials for larger employment in the fields. In this framework, a pot experiment in calcareous soil was conducted on a durum wheat variety (*Triticum Durum* L.). Irrigation with saline water (0.50 mM of NaCl for 10 days) inhibited germination, led to osmotic stress, and caused an accumulation of the toxic salt level under limited water uptake. Cell division and expansion as well as the modulation of the activity of some key enzymes were inhibited, and lastly, the utilization of the wheat reservoir was compromised. Conversely, pre-treating wheat seeds with ZnO NPs at 50 mg/L significantly improved germination rate and grain yield and attenuated the harmful effects of salinity stress. Commercial ZnO NPs with a size < 50 nm were employed in the experiment, though no further information was disclosed on the preparation or characterization of the nanomaterial [85]. The influence of seed priming with ZnO NPs and saline water irrigation on the yield and nutrient uptake of a durum wheat variety was investigated by Zaghdoud and Nagaz [86]. Also, in this case, calcareous soil was used, and seed nano-priming was carried out with 50 mg/L ZnO NPs for 4 h before sowing. The outcome of comparative treatments showed significant differences between primed and non-primed seeds, in terms of plant growth, grains yield and macronutrients uptake. In addition, ZnO NPs positively affected the growth in salt-stressed plants, stimulating the augmentation of photosynthetic pigments as well as natural auxins active in cell division. In this case commercial nanoparticles were used. The average size was reported to be <100 nm and no other information are available.

ZnO NPs were recently used as biostimulants to mitigate salt stress for *Sorghum bicolor*, an important staple food crop worldwide, widely grown in arid and semi-arid regions [84]. Plant growth was severely affected by experimental high salt stress conditions (400 mM NaCl), showing a significant decrease in shoot length and fresh weight. In order to prevent yield losses and maintain growth and production levels, sorghum seeds were primed with ZnO NPs (5 and 10 mg/L) prior to planting. The subsequent crop showed a significant increase in fresh weight under salt stress. Furthermore, by analyzing the anatomical structures, it was observed that morphological changes strictly occurred in plants grown under salt stress which could be mitigated by priming with ZnO NPs through a wider opening of the vascular bundle and an enhanced surface area of the epidermis layers responsible for the transport of water and nutrients. It was hypothesized that the action mechanism was correlated to the prevention of damage of the epidermal layers and of ion homeostasis of vascular bundle tissues [84]. The synthetic procedure of these ZnO NPs foresaw a phytochemical approach via precipitation from Zn(NO_3_)_2_ and an extract from *Agathosma betulina,* followed by freeze-drying and calcination at 600 °C, formally to improve the crystallinity of ZnO. A full characterization was carried out by multiple spectroscopic techniques, including XRD, SEM, FTIR, HRTEM, aimed at determining the properties of both the NPs and the extract. The average size of the NPs was in the 20–30 nm range.

In order to determine the efficacy of ZnO NPs priming against salinity, a study was conducted on the biological processes of two rapeseed cultivars during the early seedling stage [87]. ZnO was applied at concentrations of 25, 50 and 100 mgL^−1^, whereas NaCl was 150 mM. In general, all concentrations of ZnO NPs increased the final germination percentage and the vigor indexes. However, the highest ZnO NPs concentration assured the largest increase in shoot length, root length and biomass, with respect to control and to comparative hydropriming experiments. Nano-priming also influenced the modulation of osmotic protection, resulting in increased proline content, soluble sugar, and soluble protein contents. Furthermore, it caused a reduction in accumulated H_2_O_2_ and O_2_^−^, and chlorophyll degradation, thus improving the plant defense system. Nano-priming substituted the Na^+^ for Zn^2+^, K^+^ and Ca^2+^, and compensated the deficit of micronutrients, thus reducing the Na^+^ toxicity in the cell cytosol. Linoleic and linolenic acids were produced in larger amounts as compared to the comparative hydropriming. Moreover, the gene expression patterns of *BnCAM* and *BnPER* reflected an enhancement of germination levels, notably under the influence of ZnO 100 mgL^−1^ priming. The ZnO NPs of this investigation were synthesized by precipitation from ZnSO_4_·7H_2_O and NaOH. The precipitate was filtered, washed with distilled water, dried in a muffle furnace at a 100 °C, collected and grounded. The ensuing sample was characterized by XRD, IR, SEM and TEM, indicating the presence of polycrystalline wurtzite structure, and various functional groups and metal–oxide bonds were presented in the compound, notably, residual hydroxyls attributed to atmospheric moisture. The average size of ZnO NPs was characterized to be ~20 nm with spherical and hexagonal shapes in TEM and SEM images.

### 4.3. Heavy Metals Stress—Cu, Co, Pb and Cd

The accumulation of heavy metals in the soil environment occurs through natural and anthropogenic activities. Volcanic eruption, forest fire and sea-water spray represent natural sources of heavy metals, whereas anthropogenic activities include metal alloy processing and smelting, combustion of fossil fuels and sewage sludge [105]. Over the years, the use of industrial wastewater for agricultural irrigation led to heavy metals contamination, endangering produce and human health through the food and feed chain. Plants grown in soils rich in heavy metals absorb them through their roots, causing bioaccumulation with consequent effects on the plants’ morphology and well-being, and causing further transfer in the feed chain. Recent applications of ZnO NPs seed priming to alleviate the stress by heavy metals were conducted on the accumulation of Cu, Co, Pb and Cd.

Cu is an essential micronutrient for plant growth but is toxic if present in excessive amounts since it interferes with photosynthetic and respiratory processes, enzyme synthesis and ultrastructural development of the plants [106]. Investigations on Cu accumulations in soil where durum wheat was planted (0.50 mg/L CuSO_4_) revealed an overall inhibition of germination parameters [106]. Co-exposure of wheat seeds to CuSO_4_ and commercial ZnO NPs (average size < 100 nm) alleviated stress by decreasing Cu^2+^ ion contents, probably due to the interference of Zn^2+^ dissolved from ZnO NPs with the metabolic pathways in the plant system. This allowed an optimal level of heavy metals that does not compromise the growth and development of crops and creates a balance between antioxidant defense mechanisms and reactive oxygen species (ROS) [86]. The NPs of this investigation were purchased and used without further characterization.

The effect of ZnO NPs to mitigate cobalt uptake by crops is rarely studied. Beyond the threshold level of 50 µM, Co is considered phytotoxic. Beneficial effects were produced from priming *Zea mays* L. seeds with 500 mg/L of commercial ZnO NPs and letting them germinate under Co stress conditions (300 µM concentration). This improved plant growth, biomass and photosynthetic machinery, and reduced ROS and malondialdehyde (MDA) accumulation in maize shoots [88]. ZnO NPs decreased Co uptake and stabilized the plants’ ultra-cellular structures and photosynthetic machinery. Furthermore, a higher accumulation of nutrient content and antioxidant enzymes were found in NPs-primed seedlings, pointing at the efficient use of ZnO NPs as a stress mitigation agent for crops grown in Co-contaminated areas. The ZnO NPs of this study were characterized by XRD, SEM and TEM to determine the crystalline phase and the average size, which appeared to be in the order of 20 nm.

Pb is a persistent metal contaminant of soil, and it is toxic both to plants and to humans. In particular, Pb^2+^ can compromise cellular metabolism by replacing cations such as Ca^2+^, Mg^2+^, Fe^2+^ and Na^+^, thus interfering with their physiological functions [107]. Plants grown in Pb^2+^-contaminated soils are the principal source of human exposure, with the absorption of the metal occurring via root uptake. The weekly limit of Pb intake to avoid adverse effects to the vital organs is ~25 µg kg^−1^ of human body weight. However, this value is often exceeded in countries such as Czech Republic, China, Pakistan, Iran, Malaysia, Morocco [108]. *Basella alba* L. was selected as a target crop to investigate the potential effect of ZnO NPs seed priming to alleviate Pb toxicity [89]. A positive effect was observed in terms of an increase in seed germination, seedling and roots growth and seed vigor due Pb uptake reduction. In addition, ZnO NPs seed priming activated the plants’ defense mechanisms, increasing proline content and antioxidant enzyme efficiency, particularly during early seedling growth [89]. The evaluation of physiochemical changes in seedlings indicated an optimum ZnO NPs priming dose of 200 mg/L to reduce Pb-induced stress. The ZnO NPs employed in this investigation were synthesized at 70 °C from ZnSO_4_ and citrate and characterized by DLS and TEM and associated SAED to determine the average size (~193 nm) and estimate the crystallographic phase.

Cd is a very toxic metal that severely affects crop growth and whose concentration is increasing in arable lands. High Cd bioaccumulation generally occurs in rice plants; it is absorbed by plant roots and transferred to plant-derived foods, thus representing a health threat for populations consuming rice as a staple food [90]. Wang et al. primed two commercially popular varieties of rice seeds (Xiangyaxiangzhan and Yuxiangyouzhan), with ZnO NPs at different concentrations (0, 25, 50 and 100 mg/L), and sowed them in Cd-rich soil. No obvious effects in the seed germination of primed seeds were observed under Cd stress (*p* > 0.05); however, a substantial improvement was detected in seedling growth, root shoot length and fresh weight, as well as other related physiological parameters. In particular, the mean fresh weight of the shoot and whole seedling were increased by 17–28% upon ZnO NPs application. Modulations were observed in the superoxide dismutase (SOD) and the peroxidase (POD) activity in the roots and shoots. Furthermore, the metallothionein content in roots increased with low levels of ZnO NPs. The α-amylase and total amylase activity improved under Cd stress, and modulations in ZnO NPs uptake in the seedling were detected. Metabolomic analysis indicated that various pathways became important as rice responded to the simultaneous exposure to ZnO NPs and Cd, including the biosynthesis of phenylpropanoid and metabolism of alanine, aspartate, glutamate, taurine and hypotaurine [109]. The ZnO NPs employed in this investigation were commercial. They were characterized by SEM and TEM, indicating an average size of 36 nm and 28 nm, respectively, and matching the specifications mentioned by the manufacturer (~30 nm).

### 4.4. Arsenic Stress

Arsenic is widely present in the Earth’s crust, and it is naturally released into the environment due to geogenic effects. However, anthropogenic activities contribute to As release in soil and groundwater, causing direct and indirect exposure to humans through the consumption of contaminated food and drinking water [110]. As accumulation in the soil–plant systems interferes with metabolic plant functions, hampering seed germination and plant development, and ultimately, an overall reduction in growth indices. ZnO NPs seed priming was tested to alleviate As phytotoxicity in *Vigna mungo* (L.) Hepper plants, in both seedlings and plants before the reproductive stage [91]. Various doses of ZnO NPs were used for priming seeds (50 mg/L, 100 mg/L, 150 mg/L and 200 mg/L), that were subsequentially planted in As-treated soil. ZnO NPs seed priming significantly relieved the As stress in germination, modulating several metabolic pathways. ROS and MAD decreased whereas the accumulation of osmoregulators and the activity of antioxidant enzymes steeply increased. Seed priming mediation proved to be concentration dependent, and the best effect was observed at the highest ZnO NPs dose. Furthermore, As accumulation in the plant parts was reduced as well as the As translocation from the roots to the shoots [91]. ZnO NPs were obtained via biogenic synthesis using Zn(ac)_2_ and an extract from *vigna mungo* L. The whole synthesis was carried out at 70 °C, followed by incubation, centrifuge, drying at 60 °C and storing at 4 °C. Characterization of the NPs included UV-Vis spectra, SEM and EDX analysis to determine the average size (30–80 nm) and possible presence of impurities.

## 5. ZnO NPs Seed Priming against Biotic Stresses

Nanoparticles have been revolutionary for pest management in agriculture, since they facilitate a substantial decrease in pesticide use. At the same time, biopriming techniques, i.e., priming by microorganisms, provide alternative remedies to managing soil- and seed-borne pathogens. A proper coupling of the two approaches, such as the simultaneous application of ZnO NPs and endophytic bacteria, may ensure a sustainable crop production, improving yields and quality [111]. This was successfully achieved when facing a variety of biotic stresses that affect the yield of the most valuable pulse crops worldwide, i.e., the destructive fungus *Fusarium oxysporum*, which causes the main seedborne and soil fungus diseases in legume-growing areas [112]. In this regard, ZnO NPs were used in association with *Trichoderma harzianum,* a fungus widely employed as a biopesticide and biofertilizer, for nano-priming chickpea seeds (*Cicer arietinum* L.) [92]. Several concentrations of ZnO NPs were initially probed on the in vitro inhibition of the mycelial growth of the fungus. Based on the optimal outcome of this first phase of investigation, a concentration of 0.50 μg/mL of ZnO NPs was selected for further seed priming experiments, which caused a significant increase in the antioxidant activity of germinated plants and a decrease by 90% in the incidence of *F. oxysporum* disease. An overall increase in the defensive enzymatic activities was revealed, as well as a positive effect on plant growth, recognizing ZnO NPs as a very effective promoter agent in host tolerance under fungal stress. It must be added that *Trichoderma harzianum* was also used in the mycological synthesis of ZnO NPs. In the procedure, Zn(ac)_2_ and a filtrate of the fungus were mixed and shaken in an incubator for 24 to 48 h. Centrifugation, drying and calcination at 500 °C yielded the ZnO NPs in power form, which could easily be transported to the field for large-scale application. The NPs were characterized by XRD, FTIR, SEM, EDX and UV-Vis techniques, revealing a size in the range of 5–27 nm (TEM), and a high level of agglomeration. Furthermore, residual carbon and Al were present (determined by EDX), as well as a few functional groups, such as -NH, C-I, C-Br, C-Cl and -OH (determined by FTIR).

In addition to fungi, bacteria and parasites are major threats to legume crops. Among this sort of pathogen, *Pseudomonas syringae pv*, *pisi* and *Meloidogyne incognita* are responsible for the “bacterial blight disease complex” of pea, causing constantly increasing losses in yields [93]. The pathogens’ damage to plants may be reduced by the symbiotic action of soil bacteria with legume roots, enhanced by the associated action of the NPs. This was achieved, for instance, by exploiting the combined antimicrobial properties of *Rhizobium Frank* and ZnO NPs to pursue disease management. In vitro studies and in-pot experiments indicated a larger plant growth, chlorophyll and carotenoid content when NPs-primed seeds were grown with *R. leguminosarum* and exposed to pathogens, as compared to plants subjected to only one of the treatments. In addition, plants inoculated with *R. leguminosarum* showed higher root nodulation, whereas only a few nodules were observed in untreated plants. Both tested pathogens had adverse effects on nodulation, while the use of ZnO NPs in combination with *R. leguminosarum* reduced nodulation, blight disease indices, galling and nematode population. Finally, the segregation of various treatments in the biplot of principal component analysis demonstrated the suppressive role of ZnO NPs on the blight disease complex of pea [93]. ZnO NPs in this investigation were purchased and the average size was assessed by XRD and TEM (though the data are not shown). NPs were claimed to have a diameter in the range of 15–25 nm, and 79.1–81.5% NPs are smaller than 100 nm.

## 6. Seed Storage

During post-harvesting, seeds are prone to deteriorate under inadequate storage conditions, such as high environmental temperature and humidity. This may induce biochemical changes leading to an enrichment in protein and fat contents, and eventually causing damage and quality loss [18]. In addition, oxidative stress of proteins, lipids and nucleic acids significantly reduces germination and respiration performances. In this framework, zinc may improve the enzymatic activity in seeds, being a co-factor of several enzymes that ultimately enhance photosynthetic activity and the translocation of photoassimilates. The administration of ZnO NPs in pre-storage treatments showed great potential in seed invigoration, quality improvement and amelioration of the physiological parameters of seeds. In an investigation by Sripathy et al. [94], two different-aged lots of green gram seeds (fresh and naturally aged) were treated with ZnO NPs, and seed quality was evaluated by a standard germination test. A dose of 1000 ppm of ZnO NPs significantly increased the germination percentage in both lots of naturally aged and fresh accelerated-aged seeds. The primed seeds showed an intact seed coat, attributable to an enhanced presence of phenolic compounds, belonging to monomers of the lignin polymers [94]. Increased synthesis of ROS was observed, which was counterbalanced by the simultaneously increased activity of catalase and peroxidase. ZnO NPs were synthesized by precipitation from Zn(NO_3_)_2_ and NaOH at 55 °C and were dried at a reduced pressure at 60 °C. Subsequent characterization by SEM, EDAX, TEM and Raman suggested the formation of narrow nano-scaled rods measuring 80–150 nm in diameter, radiating from a central core.

Simultaneous storage life prolongation and action against pathogen diseases was hypothesized for chickpea seeds primed with ZnO NPs at 100 ppm and stowed in a suitable semipermeable bag. In particular, the effects of different storage lengths on plant growth and yield parameters were studied in association with the incidence of *Fusarium* wilt disease [95]. The evaluation of biochemical parameters indicated the highest activities after 6 h of seed priming, which concomitantly allowed for the prolongation of the seeds’ storage life up to 9 months. An additional action against *F. oxysporum* infection was verified, indicating the enhancement of plant growth parameters. The protection mechanism upon ZnO NPs priming was attributed to the presence of higher quantities of sugars, phenol and total proteins as well as the accumulation of superoxide dismutase (SOD) in the plants that helped to create resistance against the wilt pathogen [95]. The ZnO NPs were synthesized via a non-detailed wet chemical process, employing *Trichoderm asperellum* and characterized by UV-Vis spectroscopy, SEM, TEM and FTIR, though the corresponding spectra and images are not reported. The average size was estimated to be 33.4 nm.

## 7. Seed Priming and Germination

One of the main applications of ZnO NPs in agriculture is priming aimed at enhanced production, improved seed germination profiles and boosted seedling developments, regardless of stress sources. Recent investigations dealt with the priming of wheat (*Triticum indicum*) [96,113] by ZnO NPs at different concentrations, which produced positive effects on the germination of seeds in vitro. The optimal concentration appeared to be 20 ppm, which guaranteed a length increase in roots and shoots and a decrease in the mortality rate. A significant improvement was observed in the germination and seedling growth of *Triticum aestivum* upon treatment of wheat seeds with ZnO NPs at a concentration of 50 mg/L. The working mechanism was attributed to the enhancement and/or inhibition of parameters related to the onset of the germination processes such as the breakage of dormancy, hydrolysis, metabolization of inhibitors, imbibition and enzyme activation [81]. In a similar experiment, ZnO NPs proved to be effective in seed priming for rice (*Oryza sativa* L.), causing a reduction in germination onset and early vigor of seedlings [97]. In more detail, seed priming by ZnO NPs facilitated speedy germination by reducing the time to start emergence, the time to reach 50% germination and the mean emergence time (a measure of the rate and time spread of germination). Increased enzymatic antioxidant functions were noted that could have the function of reducing the production of degenerative radicals and protecting plants from environmental stresses during the growing stages. Seedling length (root and shoot) largely increased 18 days after sowing. All enzymatic activities were positively correlated with seed vigor indices and seedling growth upon seed priming. An overall augmentation was observed for plant chlorophyll, phenol and protein contents, leaf area index and duration, crop growth rate, uptake of nutrients (N, P, K, B, Zn and Si), and yield of direct-seeded rice. In particular, chlorophyll was associated with photosynthetic activity, and hence, it is an indicator of the vegetative growth and vigor of a plant.

ZnO NPs were synthesized according to biogenic procedures (non-detailed) and characterized by DLS, zeta potential and TEM-EDX. The mean hydrodynamic diameter of size distribution was below 10 nm, the surface charge was −5.7 mV, and the maximum intensity was at 1 keV (TEM–EDX). ZnO NPs proved to be stable in aqueous medium for up to 90 days.

Tomato is a widely spread crop with high nutritional contributions and organoleptic properties, which makes it an appetizing vegetable. The use of tomato seeds is the most accessible propagation mechanism for farmers, though the induction of germination and emergence is often hampered by the absence of stimulants promoting the development and growth of seedlings. Asmat-Campos et al. conducted a study on the influence of ZnO NPs on the germinative phase of *Lycopersicon esculentum* Mill. 1768 “tomato” with the inoculation of seeds in six sample groups (T1: Control; T2: 21.31 ppm; T3: 33.58 ppm; T4: 49.15 ppm; T5: 63.59 and T6: 99.076 ppm). The results indicated that concentrations close to 100 ppm of ZnO NPs are ideal for the promotion of enzymatic and metabolic activity to achieve cell elongation. In addition, the nanoparticles showed no phytotoxicity, as indicated by the occurrence of germination and emergence in all the treatments, which was attributed to the generation of a Zn0–phenolate complex through a chelating effect. The synthesis of the ZnO NPs was carried out using Zn(ac)_2_ ·2H_2_O and the extract of *Coriandrum sativum*, at 70 °C, followed by calcination at 500 °C. The subsequent characterization revealed the formation of ZnO nanoparticles of 30 nm in diameter with residual proteins on their surface, whose main vibrations could be located at 1384 cm^−1^ and 1494 cm^−1^, corresponding to aromatic amides and -NH in proteins, respectively [98].

Cotton is a major source of fiber for the textile and apparel industries, as well as a valuable source of protein for animal feed [99]. Its production has lowered in countries like India because of low seed germination, insect pests and diseases incidence. ZnO NPs seed priming had a positive effect on minimizing the low germinability and low vigor of American varieties of cotton seeds. Treatment with 400 ppm of ZnO NPs had the most promising effects since it increased the seed quality parameters, especially the germination percentage [114]. This was correlated to an increased production level of different enzymes such as SOD and CAT that are helpful in the germination of seeds and strengthen defense mechanisms. Furthermore, zinc acts as a promoter of the precursor of indole acetic acid that is required in auxin production, thus resulting in increased cell division. Finally, increased seed quality parameters were associated with water uptake induced by ZnO NPs during the germination process, which helps in raising healthy seedlings.

The synthesis of the ZnO NPs was carried out by precipitation from Zn(NO_3_)_2_·4H_2_O and NaOH in aqueous solution in a sealed beaker at 55 °C. The precipitate was rinsed with deionized water and ethanol and dried at 60 °C. SEM, UV-Vis spectroscopy and FTIR indicated the formation of nanoparticles of 36.25 nm in diameter carrying nitrate impurities, as demonstrated by intense IR peaks at 2358 cm^−1^ and 1541 cm^−1^, which corresponds to the C-H or C=C out-of-plane bending, related to the presence of aromatic nitro compounds.

Seed priming with ZnO NPs was probed in wheat cultivar (*Triticum aestivum H1-1544*) [100]. Comparative investigations were carried out among ZnO NPs-primed, hydroprimed and untreated seeds, resulting in a significant positive influence on seed germination performance and the vigor index of seeds nano primed with 10 mgL^−1^ of ZnO NPs. In addition, ZnO nano-priming enhanced seed water uptake, resulting in enhanced α-amylase activity and total content of photosynthetic pigments. Analysis of chlorophyll *a* fluorescence upon nano-priming, 30 days after cultivation suggested that ZnO NPs affect primary photochemistry by enhancing the performance of the water-splitting complex on the donor side of PSII (Fv/Fo). In more detail, an increase was observed in the number of active reaction centers per chlorophyll molecule, in the efficiency of excitation energy trapping, and in electron transport from active reaction centers. A prominent decrease in the activity was registered, instead, of peroxidase, catalase, superoxide dismutase and the degree of lipid peroxidation, which was attributed to low reactive oxygen species levels in nano-primed plants as compared to the control. The ZnO NPs of this investigation were commercial and characterized by the provider by TEM and XRD. The average nanoparticle size was 20–30 nm. The XRD pattern was assigned to ZnO, though unassigned spurious peaks appeared, indicating the presence of additional phases.

As an example of further developments of ZnO NPs seed priming, purposely synthesized inulin-coated ZnO NPs were proposed as biostimulants to improve the growth of *Vicia faba* L. (faba bean) seedlings [101]. To this purpose, faba beans were grown in culture medium supplemented with NPs at 50 or 100 mg kg^−1^, using five different comparative targets: ZnO NPs alone, inulin alone, a mixture of the two, or inulin-coated ZnO NPs (ZnO@inu). The investigation was carried out by evaluating, in the first instance, the seed germination rate and biometric parameters of seedlings, in association with Zn localization in the plant tissues, so that an action mechanism could be hypothesized. Additionally, photosynthetic pigments, cytotoxicity, genotoxicity and viability were carried out, along with the induction of oxidative stress and tissue damage, antioxidant response, and modulation of gene expression. The combined studies indicated the potential role of ZnO@inu NPs in the post-germinative phase, probably by stimulating stem cell mitosis. Inulin as a coating agent for ZnO NPs has a positive influence on bioavailability and adsorption into the plant tissues, without altering their bioactivity. For this investigation, naked and coated ZnO NPs were synthesized according to purposely implemented, eco-friendly protocols. In particular, a low-temperature synthetic procedure was implemented for a two-step procedure to achieve low-cost, low-sized NPs. Characterizations were carried out by XRD, FTIR, SEM and TGA analyses. It was found that ZnO NPs crystallize in the wurtzite phase, with an average size of 60 nm, which decreases to 58 nm upon coating. FTIR measurements indicated a clean ZnO surface for naked NPs, which binds the coating via the complexation of Zn atoms of the NPs. The degree of coating was also evaluated as 17,700 inulin monomeric units, or 590 polymeric units of 30 monomers each.

## 8. Challenges and Future Directions

Environmental conditions and climate changes continuously threaten crop production and endanger food quality and security. Many natural resources are being depleted and/or polluted, thus compromising human health, particularly for under-privileged communities. Over the years, the deterioration of soil and frequent water shortages due to erratic rain caused nutrient deprivation of soil and drought stress, thus reducing the crop yield. ZnO NPs nano-priming offers innovative opportunities in sustainable agriculture, enhancing crop yield and biofortification, and developing resistance against biotic and abiotic stresses. In recent years, investigations have been carried out to understand the benefits and shortcomings of ZnO NPs priming, by monitoring the modulations of the physiological and biochemical properties of treated plants. Doses and physico-chemical properties of ZnO NPs have been taken into account as primary factors in the efficacy of the procedure, indicating optimum sowing conditions and presenting new strategies for further amelioration of farming techniques. However, in order make ZnO NPs seed priming an effective and routine procedure for sustainable farming, a few key points can be identified to ensure higher crop yield and frequency, also taking into account land and climate diversities. One of the most underrepresented issues is the systematic correlation between the structure and morphology of ZnO NPs and their action as seedling and plant growth promoters. Therefore, suitable comparative studies are valued to achieve a deeper knowledge of the inherent action mechanism in correlation with ZnO NPs characteristics. Three more issues are in order to boost the efficient use of ZnO NPs in field. The first is the creation of a database where parameters are entered such as types of crop, NPs’ properties and types of soil along with their characteristics, i.e., salinity, draught level, micronutrient deficiencies, contamination and acidity, to match the most well-suited ZnO NPs stimulant. In addition, the standardization of ZnO NPs synthesis is sought for a more homogeneous and eco-friendly production. Finally, engineered ZnO NPs, with simple and low-cost coatings, are fostered to increase the production of target crops.

## Figures and Tables

**Figure 1 ijms-24-17612-f001:**
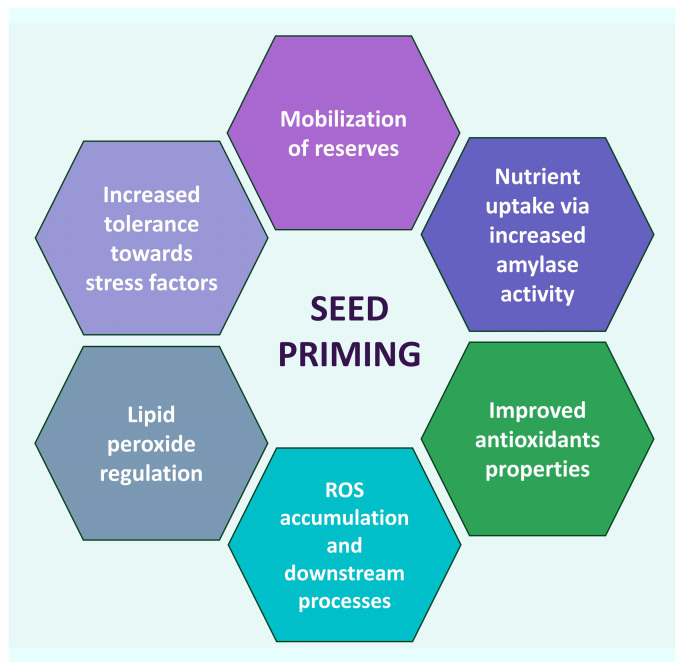
Main nano-priming mechanisms.

**Figure 2 ijms-24-17612-f002:**
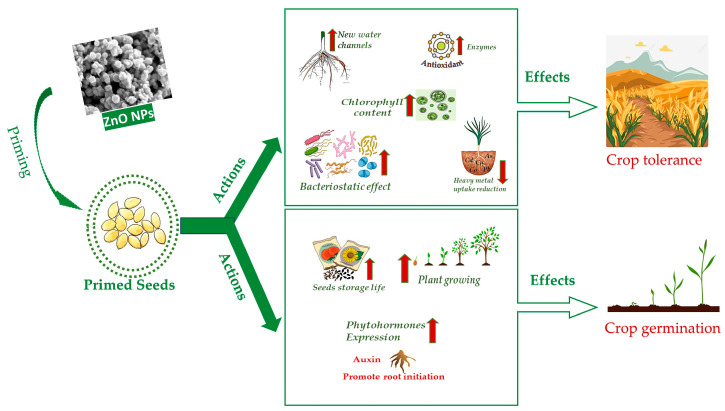
Scheme of the main actions of ZnO NPs seed priming and consequent effects on crops.

**Table 1 ijms-24-17612-t001:** Summary of the latest advances on applications of ZnO NPs seed priming, their doses and effects.

Application	Type of Seed	Doses	Effects	Ref.
Drought stress alleviation	Rice	0; 5; 10; 15; 25; 50 ppm	Agronomic profile yield	[81]
	Wheat	40; 80; 120; 160 ppm	Chlorophyll content and plant nutrients	[82]
	Wheat	10 mg/L	Reduced DNA damage	[83]
Salt stress alleviation	*Sorghum bicolor*	5; 10 mg/L	Increased Fresh weights	[84]
	Wheat	50 mg/L	Increased plant growth, grain yield and macronutrients	[85]
	Wheat	50 mg/L	Improved germination	[86]
	Rapeseed	25; 50; 100 mg/L	Increased vigor indexes	[87]
Cu stress alleviation	Wheat	20 ppm	Improved growth	[86]
Co stress alleviation	Maize	500 mg/L	Improved plant growth, biomass and photosynthetic machinery	[88]
Pb stress alleviation	*Basella alba*	200 mg/L	Increased seed germination, seedling and roots growth, seed vigor; reduced Pb uptake	[89]
Cd stress alleviation	Rice	0; 25; 50; 100 mg/L	Improved early growth and related physio-biochemical attributes	[90]
Arsenic stress alleviation	*V. mungo* (bean)	50; 100; 150; 200 mg/L	Increased germination by modulation of metabolic pathways	[91]
Biotic stress counteraction	Chickpea	0.25; 0.50; 0.75; 1.0 µg/mL 1000 mg/kg	Antifungal Activity	[92]
			Reduction in disease indices	[93]
Seed storage	*V. radiata* (bean)	1000 ppm	Increased germination percentage	[94]
	Chickpea	100 ppm	increased seed storage life; decreased disease incidence	[95]
Crop improvement	Wheat	5; 10; 15; 20 ppm	Increased plant growth	[96]
	*Oryza sativa* (rice)	10 µmol	Seedling vigor; increased plant chlorophyll content; nutrients uptake	[97]
	Tomato	100 ppm	Improved germination process	[98]
	Cotton	400 ppm	Increased germination and seed parameters	[99]
	Wheat	10 mgL^−1^	Increased germination, vigor index and chlorophyll content	[100]
	*V. faba* (bean)	50, 100 mgL^−1^	Increased biometric parameters, chlorophyll content	[101]

**Table 2 ijms-24-17612-t002:** Summary of the ZnO NPs preparation conditions and characterization, where reported.

ZnO NPs Source	Reagents and Conditions	Characterization	Size and Purity	Refs.
Purchased		Not Reported	20–30 nm 98% purity level	[81]
Purchased		UV–Vis, TEM, XRD *	Not reported—99,99%purity level	[82]
Purchased		Not Reported	Not Reported	[83]
Phyto-Synthesis	Zn(NO_3_)_2_ ·6H_2_O, *Agathosma betulina*, 80 °C, calcination 600 °C	UV–Vis, TEM, XRD, FTIR, SEM	20–30 nm	[84]
Purchased		Not reported	<50 nm	[85]
Purchased		Not reported	<100 nm	[86]
Precipitation	ZnSO_4_·7H_2_O, NaOH, drying 100 °C	XRD, IR, SEM, TEM	~20 nm	[87]
Purchased		XRD, EDX, TEM, SEM	~20 nm-99% purity	[88]
“Chemical”	ZnSO_4_, citrate, 70 °C	DLS, TEM	spherical shape ~ 193 nm High purity	[89] **
Purchased		SEM, TEM *	36 nm	[90]
Phyto-Synthesis	Zn(ac)_2_ ·2H_2_O *V. mungo* extract, 70 °C	UV–Vis, SEM, XRD	30–80 nm	[91]
Biogenic synthesis	Zn(ac)_2_ ·2H_2_O, *T. Harzanium*, 40 °C, calcination 500 °C	UV–Vis, FTIR, TEM, SEM, EDX, XRD	5–27 nm, agglomerated	[92]
Purchased		XRD	15–25 nm	[93]
Precipitation	Zn(NO_3_)_2_ ·6H_2_O, NaOH, 55 °C, vacuum 60 °C	SEM, EDAX, TEM, Raman	Narrow rods— 80–150 nm	[94]
Wet chemical synthesis	*Trichoderm asperellum*	UV–Vis, SEM, TEM, FTIR	spherical shape, 33.4 nm	[95] **
Precipitation	Zn(ac)_2_ ·2H_2_O, KOH, 60 °C, dried at 60 °C	UV-Vis, XRD, SEM, FTIR	Hexagonal shape 20 nm	[96]
Biogenic synthesis	Zn(NO_3_)_2_, ZnSO_4_, ZnCl_2_ fungi, from arid field	DLS, TEM-EDX, zeta potential	10 nm	[97] **
Phyto-Synthesis	Zn(ac)_2_ ·2H_2_O, *Coriandrum sativum*, 70 °C, calcination 500 °C	UV–Vis, FTIR, TEM, SEM, XRD	spherical shape 30 nm	[98]
Precipitation	Zn(NO_3_)_2_ ·6H_2_O, NaOH, 55 °C, Dried at 60 °C	UV –Vis, FTIR, TEM, SEM	spherical shape 36 nm	[99]
Purchased		UV –Vis, XRD, TEM	spherical shape 20–30 nm	[100]
Precipitation	Zn(NO_3_)_2_ ·6H_2_O, NaOH 40 °C–60 °C	XRD, FTIR, SEM, TGA	58–60 nm	[101]

* data provided by the manufacturer; ** and references therein.

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
