# Peer review of "Seed Priming with Zinc Oxide Nanoparticles to Enhance Crop Tolerance to Environmental Stresses"

_ijms, 2023, doi:10.3390/ijms242417612_

Round 1

Reviewer 1 Report

Comments and Suggestions for Authors

Comments on the Quality of English Language

Reviewer 2 Report

Comments and Suggestions for Authors

The review article titled "Seed Priming with ZnO Nanoparticles to Enhance Crop Tolerance to Environmental Stresses" has been reviewed, and it is my assessment that the article falls below the standard required for publication in its current form. The content and structure of the article need significant improvement to meet the criteria of a comprehensive and scientifically sound review. Below, I provide a detailed review of the article with recommendations for major revisions.

-          The title is clear, but the abstract is insufficient in providing a comprehensive overview of the article's scope and significance. It should be revised to better summarize the content and importance of the review.

-          The introduction lacks depth and fails to sufficiently establish the relevance and significance of the topic. The authors should provide a more compelling rationale for the use of ZnO nanoparticles in seed priming.

-          I could not understand section 2 results. Why this heading is the results. Give this section a some reasonable heading.

-          The literature review is inadequately organized and lacks the depth required for a comprehensive review article. There is a need for a more thorough examination of relevant studies, as well as a critical analysis of the state of the field.

-          The discussion of mechanisms is cursory, and the authors need to delve deeper into the underlying processes and provide more evidence from the literature to support their claims.

-          There should a section on “Applications in Different Crops” to discuss response of various crops to ZnO nanoparticle priming

-          Most important section of the review article “Challenges and Future Directions” is missing and it lacks specificity and clear guidance for future research. In-depth analysis and recommendations are needed.

-          The article lacks coherence and clear organization. The language used is sometimes unclear and convoluted, making it difficult for readers to follow the content.

-          The references should be reviewed and updated to include the most recent and relevant sources in the field.

-           

In summary, the article "Seed Priming with ZnO Nanoparticles to Enhance Crop Tolerance to Environmental Stresses" requires significant revisions to meet the standards expected for publication. To address these issues and make it suitable for publication, the authors should:

-           

-          Strengthen the introduction by providing a more compelling rationale.

-          Expand and reorganize the literature review to include a broader range of relevant studies.

-          Deepen the discussion of mechanisms and provide more supporting evidence.

-          Add the section on applications in different crops.

-          Add the analysis of challenges and provide more specific future directions.

-          Enhance the clarity and organization of the article.

-          Ensure that figures and tables are clear, relevant, and well-integrated.

After addressing these issues, the article may have the potential to make a meaningful contribution to the field of agricultural research and environmental stress management.

Comments on the Quality of English Language

Minor changes are required 

Reviewer 3 Report

Comments and Suggestions for Authors

The review is interesting, but has a major problem and is that is very narrow in its perspective, focusing only in a very specific point, and this limits the interest and the potential readers. Nevertheless this problem can be solved adding a little more information, which would help to contextualize and gain interest:

a) When did Zn nanoparticles started to be used? Why someone tried this technique? I have missed some historical perspective.

b) Authors focus in seed priming, but Zn nanoparticle are also used in the form of foliar spray, see for instance:

https://www.mdpi.com/2073-4395/13/1/192

I suggest to include some additional paragraphs mentioning alternative uses of Zn nanoparticles and advantages or disadvantages with seed nanopriming. 

Round 2

Reviewer 2 Report

Comments and Suggestions for Authors

Sufficiently improved and can be proceeded for the publication 

Comments on the Quality of English Language

Only a minor editing of English language required

Reviewer 3 Report

Comments and Suggestions for Authors

This version is substantially improved. I recommend publication.